# Isochromatic-Art: A Computational Dataset for Digital Photoelasticity Studies

Juan-Carlos Briñez-De-Leon [1,*], Mateo Rico-Garcia [1] and Alejandro Restrepo-Martínez [2]

1 Grupo GIIAM, Facultad de Ingeniería, Institución Universitaria Pascual Bravo, Calle 73 No. 73A-226, Medellín 050034, Colombia
2 Grupo GPIMA, Departamento de Ingeniería Mecánica, Facultad de Minas, Universidad Nacional de Colombia, Sede Medellín, Núcleo el Río, Bloque 04, Carrera 64C No. 63-120, Medellín 050034, Colombia
* Correspondence: juan.brinez@pascualbravo.edu.co

**Abstract:** The importance of evaluating the stress field of loaded structures lies in the need for identifying the forces which make them fail, redesigning their geometry to increase the mechanical resistance, or characterizing unstressed regions to remove material. In such work line, digital photoelasticity highlights with the possibility of revealing the stress information through isochromatic color fringes, and quantifying it through inverse problem strategies. However, the absence of public data with a high variety of spatial fringe distribution has limited developing new proposals which generalize the stress evaluation in a wider variety of industrial applications. This dataset shares a variated collection of stress maps and their respective representation in color fringe patterns. In this case, the data were generated following a computational strategy that emulates the circular polariscope in dark field, but assuming stress surfaces and patches derived from analytical stress models, 3D reconstructions, saliency maps, and superpositions of Gaussian surfaces. In total, two sets of '101430' raw images were separately generated for stress maps and isochromatic color fringes, respectively. This dataset can be valuable for researchers interested in characterizing the mechanical response in loaded models, engineers in computer science interested in modeling inverse problems, and scientists who work in physical phenomena such as 3D reconstruction in visible light, bubble analysis, oil surfaces, and film thickness.

**Dataset:** https://dx.doi.org/10.17632/z8yhd3sj23.5

**Dataset License:** CC BY 4.0

**Keywords:** digital photoelasticity; isochromatic images; fringe patterns; stress field; birefringence; phase maps

## 1. Introduction

In engineering applications, the stress field describes the way in which an applied force gets distributed into the geometry of a mechanical piece. The importance of its evaluation lies in the need for identifying the forces which make a piece fail, redesigning its geometry to increase the mechanical resistance, or characterizing unstressed regions to remove material. With this objective, digital photoelasticity has been widely used by many engineering areas due to its capability of being visual, non-contact, and full field, among other advantages. There, the main feature of the technique could be understood as the possibility of revealing stress information through color fringe patterns, usually called isochromatic fringes, and consequently using computational algorithms for recovering the stress values wrapped by the fringes [1,2].

Even though the digital photoelasticity has been widely applied in different engineering areas, this method is still struggling with the problem of accomplishing a successful

stress evaluation by requiring one single image acquisition, under a unique polariscope configuration [3]. There, although the literature reports different works in such direction, the absence of public data about representing the stress field through color fringe patterns has limited developing new proposals which can help to generalize the stress evaluation in a wider variety of industrial applications, and, overall, for cases with complicated fringe distributions and critical stress values [4].

Between the recent works reported in literature for digital photoelasticity, deep learning strategies are being explored, and have opened a new work line [5,6]. For that, this dataset, besides providing a big collection of photoelasticity cases, it reports essential information for allowing users to understand the phenomenological relationship between one specific stress distribution and its respective visualization through color fringe patterns overall for complicated fringe distributions that have not yet been reported in literature such as: saliency maps, 3D reconstructions, and Gaussian surfaces. This exercise is developed under a computational perspective by using analytical stress models [1,7].

Within the image generation process, this dataset considers a circular polariscope configuration guaranteeing experimental data for a PMMA birefringent material, different light sources, and a variety of camera sensors. These combinations offer to users the possibility of analyzing specific stress conditions due to a variety of isochromatic fringe representations, which is complex to obtain when working in real scientific laboratories because of the normal absence of electronic devices. In addition to the previews' experimental configurations, this dataset provides multiple stress scales for allowing users to visualize loaded pieces through different fringe densities, and the effect of dynamic load applications.

On the other hand, the fact that this dataset presents a correspondence between the stress maps and isochromatic images, as input and output, makes this work bring the opportunity of idealizing the stress evaluation process as an inverse problem, as it is usually presented in computer science applications overall in recent advances of machine learning algorithms.

The rest of the paper is presented as follows: Section 2 describes the experimental design, as well as the materials and methods used. Section 3 presents the dataset's structure and some data examples. Section 4 shows a usage example of the dataset, which consists of a convolutional neural network. Finally, Section 5 gives the conclusions.

## 2. Experimental Design, Materials, and Methods

In digital photoelasticity, the stress field refers to maps with the principal stress difference in loaded bodies, which are revealed through images with color fringe patterns by using a polariscope. In these experiments, the polariscope action could be seen as a function in which a user introduces the stressed body $\Delta\sigma = \sigma 1 - \sigma 2$, the type of light source $A(\lambda)$, and the spectral behavior of the color camera $S_{R,G,B}(\lambda)$ to obtain an image with color fringe patterns [1], as indicated by Figure 1 for a circular polariscope in a dark field [7].

Into the polariscope, the optic phenomena, besides the stress principal stress difference, requires the material properties such as the stress optic coefficient $C$ and thickness $h$. In those cases, the principal stress difference could be understood as a map with a specific intensity distribution and scaled by a defined stress magnitude. From this polariscope configuration, the emergent intensities follow Equation (1) according to the Jones calculus. This implies that generating synthetic experiments involves knowing the material optical properties $C$, body thickness $h$ with its respective map of stress distributions $\Delta\sigma$, the spectral content of light source $A(\lambda)$, and the spectral response of camera sensor $S_{R,G,B}(\lambda)$ [8]:

$$I_{R,G,B} = \frac{1}{\lambda_1 - \lambda_2} \int_{\lambda_1}^{\lambda_2} \frac{I_b(\lambda)}{2} [1 - \cos\delta(\lambda)] S_{RGB}(\lambda) \mathrm{d}\lambda \qquad (1)$$

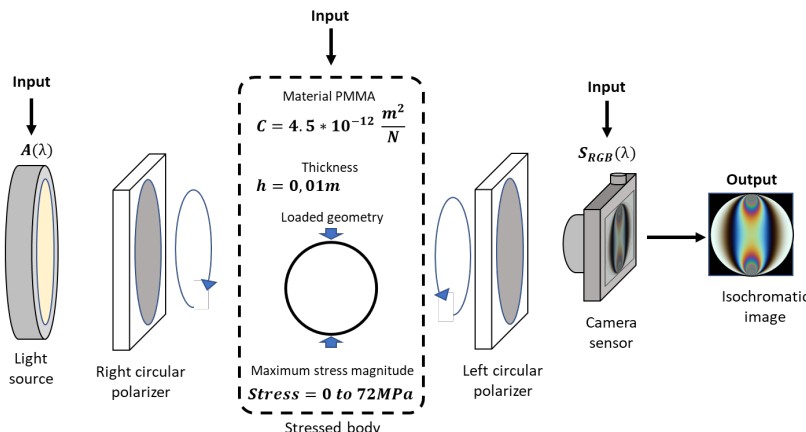

**Figure 1.** Representation of polariscope set up. Light transmission is from left to right. Here, the customizable parameters for a single image generation are marked as "Input".

In this dataset, the synthetic experiments consider real data as follows:

- Birefringent 'pmma' material of $h = 1.0 \times 10^{-3}$ mm thickness and $C = 4.5 \times 10^{-12}$ stress optic coefficient.
- Light wavelength $\lambda$ into the visible range from $\lambda 1 = 390$ nm to $\lambda 2 = 760$ nm.
- Industrial light sources $A(\lambda)$, by using discretized signals of spectral components from five different industrial devices were considered [9]. These sources are some of the most common in photoelasticity studies, such as: Constant, Incandescent, Fluorescent, and Willard_LEDGO_CN_600SC_LED; Cold white laser light is shown in Figure 2 for fluorescent and Cold white laser light.
- The remaining parameters consider stress distributions $\Delta \sigma$, and commercial camera sensors $S_{R,G,B}(\lambda)$, and combine them systematically for generating a wider variety of isochromatic images.

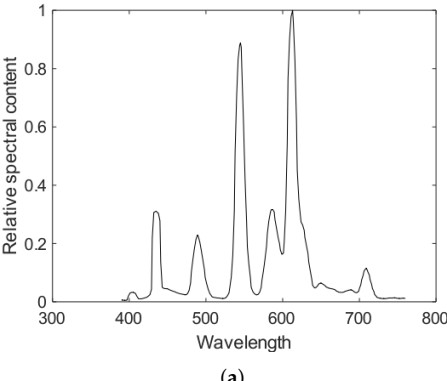

(**a**)

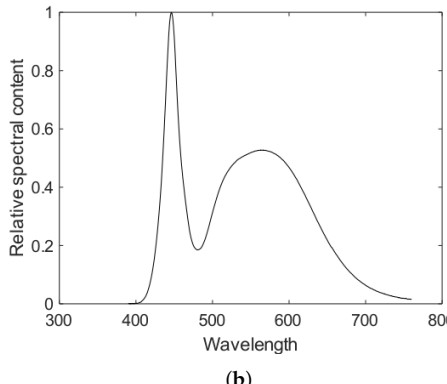

(**b**)

**Figure 2.** Relative spectral content of two light sources used in the whole dataset generation. (**a**) fluorescent; (**b**) cold white laser light.

Complementary to the light sources, this paper considers the spectral response of three types of camera sensors. In this case, the sensor signals correspond to Human vision simulation, DCC3260C, and Sony-IMX250, as illustrated by Figure 3 for human vision simulation and Sony-IMX250. For all of these sensors, the CFA effect and a demosaicing algorithms were included [1].

Finally, the last input accounts for the stress map $\Delta \sigma$. In this case, analytical surfaces are scaled to different stress magnitudes for responding to the absence of experimental data reports. In the process, the maps are resized to $224 \times 224$ pixels for matching with recent inverse problem proposals. The map generation process follows the diagram presented in Figure 4. There, the input surface is initially resized and posteriorly normalized into

a new map. In a parallel way, patches are extracted from the initial map. Then, they are resized, rotated, and normalized into new maps. At the end, all the new maps are scaled to a stress magnitude taking into account three stress values, such as: 12 MPa, 24 MPa, 36 MPa, 48 MPa, 60 MPa, and 72 MPa.

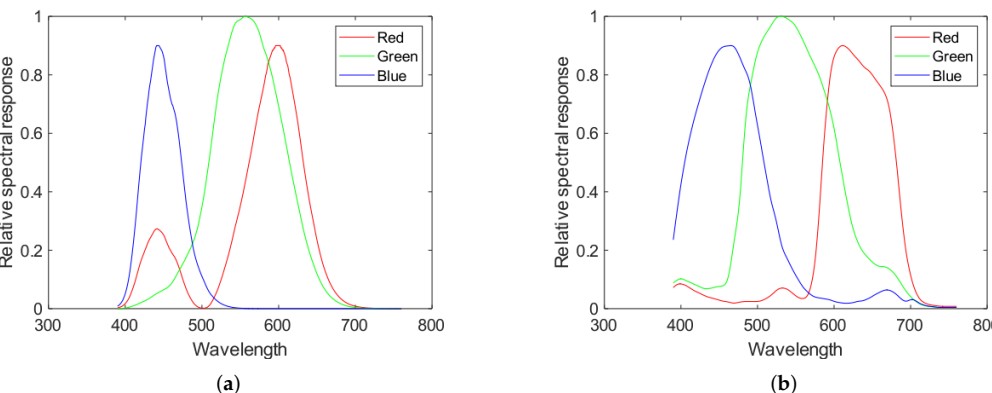

(**a**)                    (**b**)

**Figure 3.** Spectral response of two camera sensors. (**a**) simulation of human vision; (**b**) Sensor Sony-IMX250. In case of the camera's sensor, these graphs were obtained from the manufacturer data.

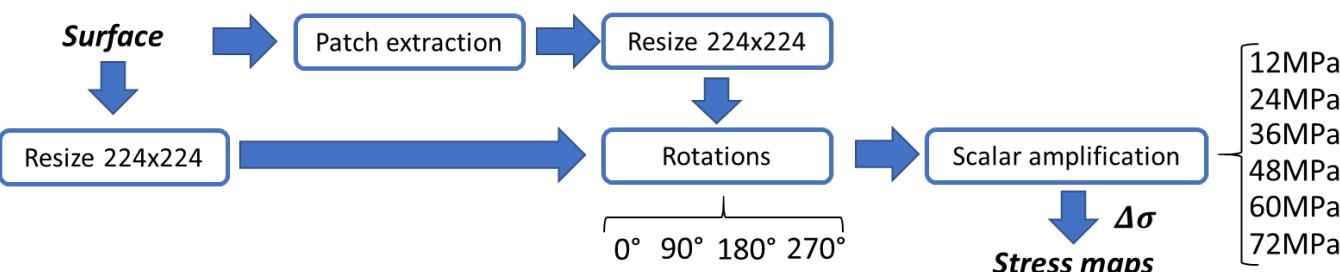

**Figure 4.** Stress map generation process. As input, a bi-dimensional surface is needed; as output of this process, a bi-dimensional stress map is generated. The load magnitude varies from 12 MPa to 72 MPa.

For a wider experimental variety, this dataset considers surfaces, normally visualized as gray images, according to four experimental cases. The first experimental case is referred to analytical stress models commonly used in photoelasticity studies [8], as summarized in Table 1 for ten conventional geometries and its respective experimental considerations. With this strategy, '4026' stress maps were sequentially obtained.

The second case starts by assuming the fixation maps in a saliency experiment reported by the IMT in the CAT2000 dataset [10], as bidimensional surfaces. In this strategy, only the fixation maps of the 'Action' clase into the trainSet are considered. With these images, '2424' stress maps were generated. With respect to the third case, this dataset generates synthetic maps taking into account random superpositions of Gaussian surfaces, as proposed for generating the benchmark in an unwrapping inverse problem [11]. With this exercise, '144' stress maps were generated having '6' surfaces as input. For the last case, the stress on the surfaces come from bidimensional representations of 3D reconstructions shared in the Stanford 3D reconstruction benchmark [12]. There, '168' stress maps were generated having '7' bidimensional representations of scanned objects as input. Chosen models are the Stanford Bunny, Armadillo, Happy Buddha, Happy Buddha Face, Lucy, and two positions of Dragon.

Posterior to the process for generating the stress maps, 15 isochromatic images were generated per every stress map according to the combinations between light sources and camera sensors. This leads to '101430' images of stress maps and the same quantity for isochromatic images.

**Table 1.** Set of geometries modeled analytically.

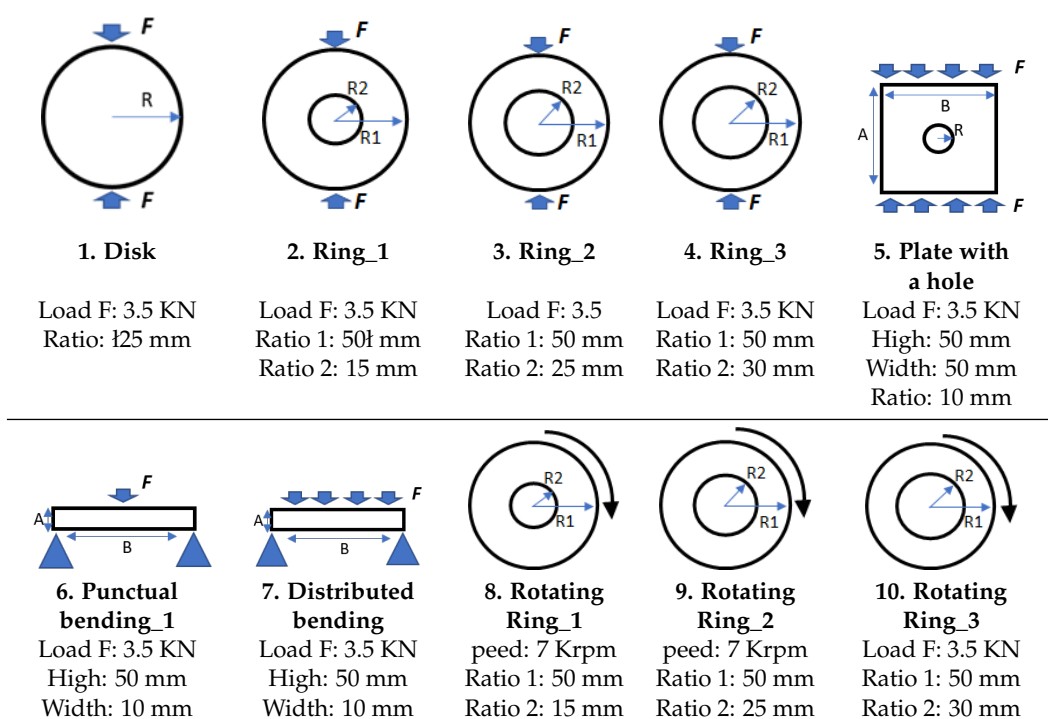

| | | | | |
|---|---|---|---|---|
| **1. Disk** | **2. Ring_1** | **3. Ring_2** | **4. Ring_3** | **5. Plate with a hole** |
| Load F: 3.5 KN | Load F: 3.5 KN | Load F: 3.5 | Load F: 3.5 KN | Load F: 3.5 KN |
| Ratio: 125 mm | Ratio 1: 50ł mm | Ratio 1: 50 mm | Ratio 1: 50 mm | High: 50 mm |
| | Ratio 2: 15 mm | Ratio 2: 25 mm | Ratio 2: 30 mm | Width: 50 mm |
| | | | | Ratio: 10 mm |
| **6. Punctual bending_1** | **7. Distributed bending** | **8. Rotating Ring_1** | **9. Rotating Ring_2** | **10. Rotating Ring_3** |
| Load F: 3.5 KN | Load F: 3.5 KN | peed: 7 Krpm | peed: 7 Krpm | Load F: 3.5 KN |
| High: 50 mm | High: 50 mm | Ratio 1: 50 mm | Ratio 1: 50 mm | Ratio 1: 50 mm |
| Width: 10 mm | Width: 10 mm | Ratio 2: 15 mm | Ratio 2: 25 mm | Ratio 2: 30 mm |

## 3. Data Example and Dataset Structure

An isochromatic-art dataset is composed of two folders of raw digital images obtained computationally from synthetic photoelasticity experiments. The first folder, called Stress maps, contains 101,430 gray images about stress maps with dimensions of 224 × 224 pixels. Names of the images in the Stress maps folder follow the next structure: 'Target' + underline + a sequential number + '.bmp'.

The second folder, called Color fringes, is composed of the images with color fringe patterns produced from the previous stress maps. Images in both folders match in quantity and dimensions. Names in the Color fringes folder follow the next structure: 'Img' + underline + a sequential number + '.bmp'.

As an example, Table 2 illustrates four experimental photoelasticity cases in the dataset. In the first case, the gray image represents the stress map about an analytical model of a ring under diametral compression whose maximum stress value is 60 MPa. The color image is the isochromatic pattern generated when observing such stress map through a circular polariscope with a fluorescent light source and a digital camera with the DCC3260c sensor.

The second case in the previous table considers a normalized fixation map of a saliency evaluation as a stress map, where maximum stress magnitude is 36 MPa. This saliency map is the '013.jpg' in the 'Train images' section of 'FIXATIONMAPS', subsection of 'Actions' into the CAT2000 dataset [10]. On the right side is the isochromatic color image generated from the left saliency map considering a circular polariscope with constant light in the visible spectral range, and a camera sensor that simulates the spectral response of the human vision.

As a third case, a normalized Gaussian surface is scaled to 24 MPa to simulate a stress map. The surface was generated by superposition of random Gaussian functions as proposed in [8]. In this case, the color image is the isochromatic representation of the Gaussian surface when considering a circular polariscope with incandescent light source, and a digital camera with the sensor Sony IMX250.

**Table 2.** Three stress maps and their respective color fringes in the Isochromatic-art dataset.

| Stress Maps Folder | | | Color Fringes Folder | |
|---|---|---|---|---|
| Case | Description | Image | Description | Image |
| 1 | Ring under compression |  | Isochromatic image for ring under compression |  |
| 2 | Saliency map |  | Isochromatic image for Saliency map |  |
| 3 | Gaussian map |  | Isochromatic image for Gaussian map |  |
| 4 | Stanford bunny |  | Isochromatic image for Stanford bunny |  |

Finally, the fourth case is a bidimensional representation in a lateral view of the Bonny Stanford 3D reconstruction [12]. This representation assumes 60 MPa as maximum stress value. In this case, the isochromatic image was generated by using a light source of laser integrations and the sensor DCC3260C in the polariscope assembly.

In addition to the image folders, this repository includes two supplemental files for a wider data understanding and reproduction.

The first file, called 'Isochromatic_art_description.xls', indicates all the experimental configurations per every generated image. There, every row is one synthetic experiment of digital photoelasticity, where the first and second columns are the names for the stress map and its isochromatic image, respectively. The third to ninth columns are the experimental case, reference model, type of abstraction, angle rotation, maximum stress magnitude, light source, and camera sensor, respectively.

The second file, called the 'stress2fringes.mat', corresponds to a Matlab$^{®}$ function used for generating the isochromatic images. In the function, the inputs are:

- Stress_Map: Continuous surface or gray map;
- Stress_Max: Maximum stress value that could exist within the experiments (in Pa);
- Stress_Magnitude: Stress magnitude to scale the continuous surface (in Pa);
- Optic_Coefficient: Stress optic coefficient (in m$^2$/N);
- Thickness: Body thickness (in m);
- Source: Data vector with the relative spectral content sampled into 371 data through the visible spectral range (371 × 1);
- Sensor: Data array with the relative spectral response of camera sensor sampled into 371 data through the visible spectral range per color component (371 × 3).

The outputs are:

- Isochromatic: 8-bit color image about isochromatic fringes with spatial dimensions according to the entered stress map;
- Stress_img: 8-bit gray image about the entered stress map.

## 4. Usage Examples: Automatic Quantification of Stress Maps from Color Fringe Patterns by Using Convolutional Neural Networks

As mentioned previously, the main intention of our dataset is to provide different study cases that can support the designing and testing of computational algorithms for demodulating the stress field wrapped by the fringe patterns in digital photoelasticity studies. To show the data validity, we propose a simple model of a convolutional neural network to predict the stress maps by using as input the isochromatic images. The model was built by stacking convolutional layers as proposed in the VGG16 model [13]. In this case, Figure 5 shows the configuration for all the layers of the proposed net. In summary, it could be said that our model can be divided into two stages: the first one to encode the fringe information, which is similar to the VGG16 architecture, and the second one to decode it into a continuous stress surface.

**Figure 5.** Illustration of the convolutional neural network architecture. The architecture is similar to VGG16 as encoder and decoder.

### 4.1. Training and Testing

Once the convolutional model was generated, the whole dataset was divided by taking into account a proportion of 80% of the cases for training, and 20% for testing. This procedure was applied to both isochromatic images, and stress maps. In the proposed model, the net takes an isochromatic image as input, and produces a stress map as output; then, the difference between the produced and the real stress map is used to improve the model leaning process. In this case, the training process was expected for 100 epochs by using the Keras toolbox in a desktop with processor Intel Xeon Silver 4208 with 16 cores to 2.10 GHz, 16 Gb RAM, and a graphic card NVIDIA QUADRO P2200.

### 4.2. Validation

Evaluating the accuracy of deep learning models used to solve an inverse problem requires the comparisons between the model output and the target expected. In our case, the recovered phase map through the net must be compared with the reference phase map. To this, there exist different similarity metrics to carry out such comparisons. However, some of the most reported are the mean squared error (MSE) [14], peak signal-to-noise ratio (PSNR) [15], and the structural similarity index (SSIM) [16]. In the MSE case, the metric is based on the squared differences between the spatial magnitudes of the reference map $\delta_{ref}$, and the magnitude of in the map generated with a convolutional model $\delta_{pred}$, as it is

indicated in Equation (2). There, values close to zero indicate high similarity between tithe maps, and, therefore, high performance of the model.

$$MSE = \frac{1}{M \times N} \sum_{x=1}^{M} \sum_{y=1}^{N} [\delta_{ref}(x,y) - \delta_{pred}(x,y)]^2 \tag{2}$$

Complementary to the MSE, the PSNR error evaluates the similitude between the phase maps but introduces a relation with the maximum phase value into the reference case, as it is indicated in Equation (3). This operation allows the metric to be more specific to measure the quality of reconstruction tasks. In this metric, high values indicate better performance; on the contrary, low values indicate low performance:

$$PSNR = 10 \log_{10} \left( \frac{max(\delta_{ref})}{MSE} \right) \tag{3}$$

In the case of the SSIM, the metric was designed exclusively for comparison of images. The SSIM measures the quality in a predicted image, phase map in our case, from the reference information by analyzing contrast, luminance, and structure, as it is indicated in Equation (4). There, 'C1' and 'C2' are compensation coefficients that depend on the dynamic range. In that case, $\delta$ accounts for the co-variance. SSIM results close to '1' indicate high performance, results close to '0' indicate low performance:

$$SSIM = \frac{(2 * \delta_{ref} * \delta_{pred} + C1) * (2 * \sigma + C2)}{(\delta_{ref}^2 + \delta_{pred}^2 + C1) * (\sigma_{ref}^2 + \sigma_{pred}^2 + C2)} \tag{4}$$

Among the variety of results that were obtained within this validation process, Table 3 shows some highlighted cases. There, the values of the metrics between the target and prediction data confirm the advantages of using the proposed dataset when developing machine learning algorithms for quantifying the stress field from a single isochromatic image. In the table, although some cases indicate low prediction performance, these explorations open a new horizon of works due to the need to achieve successful demodulation of the fringe patterns when having photoelasticity cases that produce fringe patterns with variations of illumination, distribution, and density.

**Table 3.** Predicted stress maps from isochromatic images.

| | Fringes | Target | Predicted | Metrics |
|---|---|---|---|---|
| Disk patch |  |  |  | MSE: 25.73 PSNR: 35.54 SSIM: 0.9750 |
| Ring patch 1 |  |  |  | MSE: 127.02 PSNR: 29.12 SSIM: 0.9454 |
| Ring patch 2 |  |  |  | MSE: 28.16 PSNR: 34.38 SSIM: 0.6338 |

**Table 3.** *Cont.*

| | Fringes | Target | Predicted | Metrics |
|---|---|---|---|---|
| Complete Ring |  |  |  | MSE: 75.86<br>PSNR: 34.27<br>SSIM: 0.8476 |
| Bunny |  |  |  | MSE: 150.47<br>PSNR: 31.85<br>SSIM: 0.9042 |
| Ring |  |  |  | MSE: 129.76<br>PSNR: 33.33<br>SSIM: 0.8912 |

## 5. Conclusions

A computational dataset for isochromatic images, and stress maps in digital photoelasticity studies, were reported in this paper. With this work, we have provided a big collection of photoelasticity cases from different experimental configurations, which are difficult to obtain when working with real experimental scenarios. With these images, researchers in digital photoelasticity will be capable of making fast validations of existing techniques requiring one single-acquisition. Likewise, this dataset will open up the opportunity to explore new strategies for fringe pattern demodulation by implementing machine learning algorithms due to the view of computational strategies for inverse problems that take as input images with isochromatic fringe patterns. In addition, expect as output the demodulated stress map.

Different to the conventional images reported in photoelasticity studies, this dataset provides variability in fringe distributions and density. In our case, we have reported synthetic isochromatic images generated computationally by considering different experimental conditions such as: complicated stress distribution, load increments, light sources, camera sensors, noise effects, rotations, and patch extractions.

Although the proposed dataset was based on computational procedures, we believe that some of the most valuable features include the possibility of providing examples for a better understanding of the photoelasticity method and the birefringent phenomenon. Another important feature is the possibility of carrying out fast explorations of the fringe demodulation process by using modern models of machine learning techniques, and, after that, the possibility of making an appropriate tuning process with different types of images to be used in any engineering application.

Between the contributions generated by this dataset, some are listed as follows:

- For researchers in different engineering areas, the possibility of having a big collection of isochromatic images, and its respective stress maps, open up the opportunity to explore new strategies for bi-dimensional inverse problems. This implies that the set of images becomes a chance for amplifying the range of applications based on deep learning methods overall for such optical areas where obtaining data comes from complicated experiments.
- In areas related to holography, a giant set of phase information, visualized through color fringe patterns, allows researchers in different engineering areas to explore unwrapping processes where the fringes adopted complicated distributions, appearance, and concentrations, which is difficult to obtain experimentally.

- In photoelasticity studies, besides providing a big variety of experimental effects, which are not possible physically due to the absence of geometries, type of loads, light sources, camera sensors, resolution, etc.; our dataset allows for defining benchmarks that afford the comparisons between the conventional and non-conventional methods, such as those new reports supported by deep learning algorithms.
- Finally, with this dataset, our future work is focused on developing and tuning different architectures in deep learning models for demodulating the stress field of loaded pieces while acquiring a single isochromatic image.The idea in this proposal lies in the opportunity of using our dataset for testing new models reported in computer science for deep learning, and achieving a successful unwrapping process of the stress map.

**Supplementary Materials:** The following supporting information can be downloaded at: https://data.mendeley.com/datasets/z8yhd3sj23/5.

**Author Contributions:** Conceptualization, J.-C.B.-D.-L. and A.R.-M.; methodology, J.-C.B.-D.-L. and A.R.-M.; software, J.-C.B.-D.-L. and M.R.-G.; validation, J.-C.B.-D.-L., A.R.-M. and M.R.-G.; formal analysis, J.-C.B.-D.-L., A.R.-M. and M.R.-G.; investigation, J.-C.B.-D.-L., A.R.-M. and M.R.-G.; resources, J.-C.B.-D.-L.; data curation, J.-C.B.-D.-L., A.R.-M. and M.R.-G.; writing—original draft preparation, J.-C.B.-D.-L. and M.R.-G.; writing—review and editing, J.-C.B.-D.-L. and A.R.-M.; visualization, J.-C.B.-D.-L. and M.R.-G.; supervision, J.-C.B.-D.-L. and A.R.-M.; project administration, J.-C.B.-D.-L.; funding acquisition, J.-C.B.-D.-L. All authors have read and agreed to the published version of the manuscript.

**Funding:** This research received no external funding. The APC was funded by Instución Universitaria Pascual Bravo.

**Informed Consent Statement:** Not applicable.

**Data Availability Statement:** Briñez-de León, Juan Carlos; Rico-García, Mateo; Restrepo-Martínez, Alejandro (2022), "Isochromatic-art: a computational dataset for digital photoelasticity studies.", Mendeley Data, v5, DOI: https://dx.doi.org/10.17632/z8yhd3sj23.5.

**Acknowledgments:** Special thanks to the projects 'RI201902', and 'IN202009' at the Institución Universitaria Pascual Bravo. We also thank the '647' Colciencias scholarship, the Facultad de Ingeniería at the Institución Universitaria Pascual Bravo, Mechanical Engineering department, and the System Engineering department at the Universidad Nacional de Colombia–Sede Medellín.

**Conflicts of Interest:** The authors declare no conflict of interest.

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
