# Peer review of "Isochromatic-Art: A Computational Dataset for Digital Photoelasticity Studies"

_data_

Round 1
Reviewer 1 Report
A digital photoelasticity dataset was generated. The dataset considers a circular polariscope configuration warrantying experimental data for a PMMA birefringent material, 5 different light sources and a variety of camera sensors.
This is an excellent dataset for digital photoelasticity based on deep learning.
Author Response
We thank the editor and reviewers for their comments aimed at improving the quality of our paper ”Isochromatic-art: a computational dataset for digital photoelasticity studies” (data-1848745). In the attached document we address their concerns point by point.

Reviewer 2 Report
Article is very interesting for academia.
Author Response

(The authors gave the same response as above.)

Reviewer 3 Report
This paper provided two sets of images for isochromatic color fringe and images generated for stress maps. I think this data is deserved to highlight because it is very difficult to find a sufficient amount of well-organized raw images to be used in recent machine learning technologies. This study is very interesting, and the paper is logically written and well organized. I would like to recommend to publish this paper in the Data journal.
Author Response

(The authors gave the same response as above.)
